# Oral Microbial Translocation Genes in Gastrointestinal Cancers: Insights from Metagenomic Analysis

**DOI:** 10.3390/microorganisms12102086

**Published:** 2024-10-18

**Authors:** Linqi Wang, Qinyu Wang, Yan Zhou

**Affiliations:** State Key Laboratory of Genetic Engineering, School of Life Sciences, Fudan University, Shanghai 200438, China; linqiwang21@m.fudan.edu.cn (L.W.); qinyuwang22@m.fudan.edu.cn (Q.W.)

**Keywords:** metagenomics, oral pathogen, colorectal cancer, inflammatory bowel disease, oral-gut axis

## Abstract

Along with affecting oral health, oral microbial communities may also be endogenously translocated to the gut, thereby mediating the development of a range of malignancies in that habitat. While species-level studies have proven the capability of oral pathogens to migrate to the intestine, genetic evidence supporting this mechanism remains insufficient. In this study, we identified over 55,000 oral translocation genes (OTGs) associated with colorectal cancer (CRC) and inflammatory bowel disease (IBD). These genes are primarily involved in signal transduction and cell wall biosynthesis and show consistency in their functions between IBD and CRC. Furthermore, we found that *Leclercia adecarboxylata*, a newly discovered opportunistic pathogen, has a significantly high abundance in the gut microbiota of colorectal cancer patients. OTGs of this pathogen were enriched in 15 metabolic pathways, including those associated with amino acid and cofactor metabolism. These findings, for the first time, provide evidence at the genetic level of the transfer of oral pathogens to the intestine and offer new insights into the understanding of the roles of oral pathogens in the development of gastrointestinal cancers.

## 1. Introduction

The intestine serves as the largest, most complex, and well-studied microbial ecosystem in the human body. Its diverse microbiota contributes to regulating and maintaining host physiology, including host immunity, nutrient digestion, and defenses against colonization by pathogenic microorganisms [1,2,3]. Given the important impact of the gut microbiota, disorders of the gut microbiota have been shown to underlie a variety of gastrointestinal pathologies, including inflammatory bowel disease (IBD) [4,5] and colorectal cancer (CRC) [6]. IBD is a chronic, non-specific inflammatory disorder that consists primarily of two subtypes: Crohn’s disease (CD) and ulcerative colitis (UC). IBD is the most definitive risk factor for the onset and progression of CRC, and thus the two diseases may share common etiologic factors in their pathogenesis, including marked changes in the microbiome [7]. There is evidence suggesting that the gut microbiota is closely associated with the occurrence and progression of colorectal cancer, and the characteristics of the gut microbiota appear to be influenced by factors such as age, diet, geographical location, family history, body mass index (BMI), etc. [8,9,10,11]. Some researchers have proposed diagnostic tests based on gut microbiota (biomarkers), which can reliably and accurately identify risk factors for CRC and IBD [12,13,14,15].

Recent studies have reported abnormal enrichment of several oral resident bacteria, such as *Fusobacterium* spp., *Klebsiella* spp., *Streptococcus* spp., *Porphyromonas* spp., *Prevotella* spp., and *Staphylococcus* spp. in patients with IBD [16,17,18] and CRC [19,20,21]. Currently, there is no evidence to suggest that the oral microbiota is a primary cause of IBD or CRC. One hypothesis is that periodontitis leads to the expansion of oral pathogens, such as *Porphyromonas gingivalis* and *Fusobacterium nucleatum*, whose migration can disrupt gut microbial community structures and trigger intestinal inflammation [22,23]. Another hypothesis posits that periodontitis induces the production of Th17 cells reactive to oral pathogens, which exhibit gut tropism. Upon migrating to inflamed intestines, these Th17 cells, when stimulated by oral pathogens, can promote the onset of colitis [24]. Other studies have pointed out that individuals with gastrointestinal diseases demonstrate higher rates of oral–intestinal bacterial transmission [25,26], suggesting that compromised gastrointestinal barriers in pathological states may accelerate the translocation and colonization of oral pathogens.

However, which oral bacteria are involved in the translocation and colonization of the gut remains controversial [16,27]. Most studies have used 16S rRNA gene sequencing technology to explore the composition and abundance of gut microbiota, with taxonomic resolution often limited to the genus level. Rapidly advanced shotgun sequencing technology in the past several years has enabled taxonomic discrimination at higher resolution, and meta-analyses across cohorts have further enhanced comparability among different studies [21,25,28]. Additionally, although there was evidence suggesting the translocation of oral pathogens into the gut, the involvement of oral pathogens in the development of gastrointestinal cancers remains inadequately understood. Microbial genes are a valuable unit of analysis because they can be linked to microbial genomes, taxonomic annotations, and predicted metabolic functions, leading to a comprehensive understanding of genetic characteristics and functional potential. Several well-studied oral pathogens have been shown to interact with gastrointestinal or immune cells through specific genetically encoded molecules, which in turn affect the integrity of the gastrointestinal barrier and the progression of gastrointestinal inflammation [22,29]. Therefore, the identification of links between the oral and gut microbiota at the genetic level may provide a novel perspective to understand the relationship of gut microbiota-related diseases.

In this work, we explore for the first time the genetic relationship between gastrointestinal cancers and pathogenic oral bacterial translocations using cross-cohort metagenomic data. We identified over 55,000 oral translocation genes (OTGs) associated with IBD and CRC using multiple differential gene analysis methods. The distribution and function of these genes were then analyzed in various species, as well as the metabolic pathways in which they are involved. We also compared the functional similarities and differences of these translocate genes in two types of gastrointestinal cancers, which provides new ideas for studying the role of oral pathogenic bacteria in the development of gastrointestinal cancers.

## 2. Materials and Methods

### 2.1. Data Acquisition

Oral metagenomic sequencing data and related metadata were downloaded from the Human Microbiome Project Data Portal (https://portal.hmpdacc.org/ (accessed on 30 October 2022)). Samples meeting the requirements for whole-genome sequencing and data size ≥ 4 GB were retained for further analysis, while samples affected by drug effects were excluded from the dataset. Sequencing data of samples from the intestines and other body sites were also downloaded from public databases, including the Sequence Read Archive (SRA, https://trace.ncbi.nlm.nih.gov/Traces/sra/ (accessed on 16 November 2022)), the European Nucleotide Archive (ENA, https://www.ebi.ac.uk/ (accessed on 16 November 2022)), and the DNA Data Bank of Japan (DDBJ, https://www.ddbj.nig.ac.jp/ddbj/index-e.html (accessed on 16 November 2022)). The same screening criteria were implemented as with oral samples.

### 2.2. Data Pre-Processing and Sample Filtering

Metagenomic reads were initially processed using fastp (v0.23.1) [30] to trim sequences with more than 3 N bases or those shorter than 60 bases. Subsequently, the reads underwent alignment to the human genome chm13.draft_v1.0_plusY via Bowtie2 (v2.3.5.1) [31] with default settings to remove host contamination. Samples that did not meet the quality standards with a Q30 ≤ 85%, a duplication rate ≥ 30%, or a contamination rate ≥ 5% were excluded from subsequent analyses. Detailed sample information is provided in Table 1 and Appendix A.

### 2.3. Taxonomic Profiling and Principal Component Analysis

Taxonomic profiles of each sample were obtained by running MetaPhlAn3 [47] with the default settings. Subsequently, we performed a centered log-ratio (CLR) transformation of the species relative abundance data followed by principal component analysis (PCA) using R. Statistical (v4.2.3) significance among multiple groups was analyzed by PERMANOVA. A *p*-value of <0.05 is considered statistically significant. Outliers were determined using Hotelling’s *T*-squared distribution and the squared prediction error (SPE/DmodX) detection methods.

### 2.4. Robustness Test of HUMAnN3 with Simulated Data

A simulation dataset of bacterial genomes was generated by randomly selecting 100 genomes from the ChocoPhlAn3 [47] database. The simulated dataset was then treated with InSilicoSeq (v1.5.3) [48], a sequencing read simulator, and ten paired-end sequencing files in FASTQ format (Illumina, 250 bp*2) with a size of 50 Mb were generated. Afterward, HUMAnN3 [47] was run with various combinations of the “--prescreen-threshold parameter” (0.01, 0.001, and 0.0001) and the “--nucleotide-subject-coverage-threshold” (0, 25, 50, 75, 100). The numbers of true positive (TP) genes, false positive (FP) genes, and false negative genes in the test, the ratio of reads aligned to the simulated genome, and the running time of HUMAnN3 were used to evaluate the effectiveness. The precision was calculated as TP/(TP + FP), and the sensitivity, also known as recall, was calculated as TP/(TP + FN). The prescreen-threshold parameter mainly affects the running time of the program, but has little effect on accuracy, sensitivity, and alignment rates (Appendix A). This result is predictable as lower genome coverage thresholds incorporate a larger set of pan-genomes, thereby extending the alignment time. The nucleotide-subject-coverage-threshold parameter, on the other hand, demonstrated higher sensitivity. HUMAnN3 achieves a precision above 97.5% and a recall near 87.5% when gene coverage is set below 50. Although precision remains high, recall drops to 80% when this threshold is raised to 75. Notably, recall and alignment rates sharply decline to nearly 0 when gene coverage is adjusted to preserve only sequences that match the entire gene length (i.e., nucleotide-subject-coverage-threshold = 100). Therefore, we consider the prescreen-threshold = 0.0001 and nucleotide-subject-coverage-threshold = 25 as the optimal parameter combination to obtain high precision, recall, and alignment rates while keeping the runtime within a reasonable range.

### 2.5. Identification and Filtering of Oral Translocation Genes

Considering the basal bacterial abundance differences between the oral and gut environments, we aimed to investigate translocation events directly at the gene level. Candidate translocation genes (OTGs) were defined as those that did not exhibit significant differences in expression between oral and gut samples in disease contexts yet demonstrated significant differences in expression between oral and gut samples in healthy contexts, as well as between healthy and disease samples in the gastrointestinal tract. Three statistical methods were used to identify genes associated with translocation events, including the nonparametric Wilcoxon rank-sum test [49], a widely used differential expression analysis tool DESeq2 [50], and MetagenomeSeq [51], a method for detecting differential abundance of microbes in metagenomic data. Benjamini–Hochberg’s false discovery rate (BH-FDR) [52] was applied for the correction of multiple testing. Genes that consistently met these criteria across two or more statistical methods were ultimately identified as OTGs associated with each disease.

### 2.6. Functional Annotation and Pathway Enrichment

The latest release of the Clusters of Orthologous Genes (COG) database (COG-20) was downloaded from https://ftp.ncbi.nih.gov/pub/COG (accessed on 8 October 2023). Translocation gene sequences were aligned against the COG database using BLAST (v2.2.26) [53] with default parameters for annotation. KEGG pathway analysis and enrichment were performed on translocation genes using an online tool KEGG Orthology Based Annotation System (KOBAS (v3.0.3)) [54]. When multiple genomes are available for a species, the one with the most complete assembly is chosen as the reference for enrichment analysis.

## 3. Results

### 3.1. Comparison of Microbial Composition across Body Sites and Disease Status

A total of 363 oral samples and 1661 gut samples related to IBD/CRC were collected from 14 public datasets across the globe. To assess potential contamination of gastrointestinal and oral samples by microorganisms from other body parts, we also collected 881 representative metagenomic samples from the respiratory tract, skin, and vagina (Table 1). All shotgun sequencing data of human samples were processed with MetaPhlAn3 for quantitative species-level taxonomic profiling (Appendix A), and the microbiota composition of samples from various body sites was evaluated by principal component analysis (Figure 1). Figure 1A revealed that the heterogeneity among different body sites had a significant effect (PERMANOVA R^2^ = 0.113, *p*-value = 0.001) on the microbial composition. After removing four outliers in the gastrointestinal and oral samples, the first two principal components explained 36.03% of the total variance, and the between-group variance value decreased to 0.110. Pairwise comparisons between the body sites, including the gastrointestinal and oral samples, are detailed in Appendix A, showing significant differences in microbial composition between these body sites. We also noticed that some of the samples from different body parts were close to or overlapped with each other, suggesting that there may be microbial translocation or colonization in different body parts. To further explore the relationship between oral microbiota and gut disorders, we performed independent PCA analyses on healthy oral samples as well as gut samples with colorectal cancer, Crohn’s disease, and ulcerative colitis (Figure 1B). However, no overlap between gut and oral samples was observed in PCA plots, suggesting that the translocation of oral bacteria in healthy individuals may have a limited effect on the gut microbial composition. Nonetheless, this finding prompts the consideration of the hypothesis that microbial translocation between the oral cavity and the digestive tract may still occur in healthy individuals. Furthermore, this exchange could also take place in the opposite direction, particularly in instances of regurgitation or mild gastrointestinal inflammation.

### 3.2. Comparison of Differential Abundance Analysis Methods in Identifying Translocation Genes

We defined candidate translocation genes (OTGs) based on expression patterns observed in both oral and gut samples. Specifically, OTGs were characterized as genes that exhibited no significant expression differences between oral and gut samples in disease contexts yet showed notable differences in healthy contexts. Furthermore, these genes displayed significant expression variations between healthy and disease states within the gastrointestinal tract. To identify these OTGs, we employed three statistical methods (Figure 2). For CRC samples, the classical nonparametric test, Wilcoxon [49], identifies the fewest translocation genes. In contrast, DESeq2, renowned for its high sensitivity [50], uncovers more than five times the number of differential genes compared to Wilcoxon by searching for genes with significant differences between groups using a model based on a negative binomial distribution. MetagenomeSeq is a differential analysis method designed for metagenomic data that addresses the effects of zero inflation and inter-sample heterogeneity [51] and reveals a number of translocation genes comparable to those detected by DESeq2. For IBD-CD samples, MetagenomeSeq identified a greater number of translocate genes compared to Wilcoxon and DESeq2, with 2.7 and 3.7 times more genes detected, respectively. For IBD-UC samples, on the other hand, Wilcoxon and MetagenomeSeq showed similar results, with 1.4 times the number of translocation genes than DESeq2. Considering the strengths of each method, genes detected in two or more differential abundance analysis methods were filtered as reliable OTGs. Therefore, the final numbers of OTGs in CRC, IBD-CD, and IBD-UC were 28,992, 13,408, and 21,028, respectively.

### 3.3. Functional Profiling and Pathway Enrichment of OTGs in CRC

Functional signatures of OTGs in CRC were described by the Clusters of Orthologous Groups of proteins (COG) annotation and the Kyoto Encyclopedia of Genes and Genomes (KEGG) pathway enrichment. The top five functional classifications with the most OTGs, [K] transcription, [G] carbohydrate transport and metabolism, [M] cell wall/membrane/envelope biogenesis, [E] amino acid transport and metabolism, and [L] replication, recombination, and repair, accounted for 38.4% of the total number of OTGs (Appendix A). Few OTGs were found in [A] RNA processing and modification, [B] chromatin structure and dynamics, [Y] nuclear structure, and [Z] cytoskeleton, suggesting a specific role of OTGs in cellular functions and metabolism (Appendix A).

OTGs in CRC were enriched in the KEGG pathways of 14 species (Appendix A). Genes from *Bifidobacterium animalis* (*B. animalis*) were enriched in the ribosome pathway, suggesting the potential influence of this bacteria on gut microbiota stability by impacting protein synthesis processes. *Gemella morbillorum* (*G. morbillorum*) was reported as an oral resident pathogen that is associated with CRC [28]. Our results showed that genes from this pathogen were enriched in metabolic pathways, synthesis of secondary metabolites, and peptidoglycan biosynthesis, providing additional evidence of its potential role in gut disorders. Notably, as a rare human pathogen commonly found in immunosuppressed hosts [55], *Leclercia adecarboxylata* (*L. adecarboxylata*) had 15 enriched KEGG pathways, including biosynthesis of secondary metabolites and amino acid, microbial metabolism in diverse environments, and pathway of various metabolisms (Figure 3). We also found that the second-level KEGG pathways of global and overview maps and carbohydrate metabolism had the most enriched third-level pathways involving various important biochemical responses and regulatory mechanisms. These pathways might reflect the adaptive changes and functional differences of different species during CRC development.

We further identified differentially expressed OTGs in the gut microbiota between CRC patients and healthy controls (Appendix A). In total, 1865 genes were upregulated and 3127 were downregulated, representing 5.9% and 9.9% of the total OTGs, respectively. The most pronounced log2 fold changes were observed in the top 10 upregulated and downregulated genes (Appendix A). Among the upregulated genes, many were associated with *Prevotella copri* and *Elizabethkingia bruuniana*. In contrast, the downregulated genes were predominantly from *Parvimonas micra*, suggesting its possible involvement in CRC-associated gut dysbiosis.

### 3.4. Comparison of Functional Profiling and Pathway Enrichment of OTGs in CD and UC

To elucidate the genetic distinctions of oral bacterial translocation events in Crohn’s disease (CD) and ulcerative colitis (UC), we compared the functional classifications and pathway enrichment patterns of OTGs in these two subtypes of IBD (Appendix A). The results showed that the top five functional classifications with the most OTGs were consistent in CD and UC, including [J] translation, ribosomal structure, and biogenesis, [G] carbohydrate transport and metabolism, [M] cell wall/membrane/envelope biogenesis, [K] transcription, and [E] amino acid transport and metabolism (Appendix A). Similar to the results in CRC, there were almost no OTGs associated with [A] RNA processing and modification, [B] chromatin structure and dynamics, [Y] nuclear structure, and [Z] cytoskeleton, as well as [W] extracellular structures. These findings emphasize the commonalities between CD and UC at both the genetic and functional classification levels and also reflect the common features between IBD and CRC.

Significantly enriched pathways for OTGs in CD were found in eight species (Figure 4 and Appendix A). The enriched pathways for OTGs of *Actinomyces* sp. oral taxon 414 and *Bifidobacterium longum* (*B. longum*) were ribosome and ABC transporters, respectively, indicating their activity in protein synthesis and material transport. Both *G. morbillorum* and *Streptococcus mitis* (*S. mitis*) showed pathway enrichment in teichoic acid biosynthesis. However, two other species of *Streptococcus* spp., *Streptococcus oralis* (*S. oralis*) and *Streptococcus salivarius* (*S. salivarius*), had OTGs enriched in quorum sensing, indicative of their capabilities in bacterial communication and coordination. Two species of *Veillonella* spp. also showed different enrichment features: *Veillonella parvula* (*V. parvula*) had the most OTGs in CD, primarily enriched in metabolic pathways, as well as the biosynthesis of secondary metabolites, cofactors, and amino acids. *Veillonella rodentium* (*V. rodentium*), however, had more OTGs enriched in microbial metabolism in diverse environments, along with metabolic pathways of carbon, propanoate, glyoxylate, and dicarboxylate. Five species in UC had significantly enriched KEGG pathways, four of which were also found in CD, underscoring their significance in IBD (Appendix A). Two species, *S. mitis* and *S. oralis,* showed consistent enrichment patterns with those observed in CD, while OTGs of *Actinomyces* sp. oral taxon 414 and *S. salivarius* transitioned to enrichment in the oxidative phosphorylation and two-component system, respectively. OTGs of *Megamonas hypermegale* (*M. hypermegale*) were significantly enriched in the pathway of ABC transporters, aligning with the enrichment characteristics observed in the *B. longum* species of CD. The most frequently enriched pathways for OTGs in IBD were quorum sensing, teichoic acid biosynthesis, and ABC transporters. Different species of *Veillonella* spp. and *Streptococcus* spp. had a substantial number of OTGs as well as enriched pathways, emphasizing their potentially significant roles in the development of IBD.

In Crohn’s disease, we observed significant upregulation of several OTGs, particularly from *Gemmiger formicilis* and *Eubacterium ventriosum* (Appendix A). Additionally, *Faecalibacterium prausnitzii*, known for its anti-inflammatory properties, was also upregulated, which is consistent with its association with gut health. The most significantly downregulated OTGs belonged predominantly to *Veillonella parvula*, suggesting its possible involvement in the dysbiosis linked to Crohn’s disease. For ulcerative colitis, the upregulated OTGs were primarily associated with *Prevotella copri* and *Bacteroides plebeius*, both of which are known to play roles in carbohydrate metabolism and immune response (Appendix A). *Megamonas funiformis* was also among the upregulated species, potentially highlighting its involvement in the pathophysiology of UC. On the other hand, the downregulated genes were mainly linked to *Actinomyces* and *Streptococcus anginosus*, indicating their relevance in UC-related microbial shifts.

### 3.5. Common Features of OTGs in CRC and IBD

Analyses of oral translocation genes (OTGs) in colorectal cancer (CRC) and inflammatory bowel disease (IBD) offer valuable insights into these two conditions and underscore the correlations of OTGs in terms of metabolic pathways and biological functions. *S. oralis* was found to carry a large number of OTGs both in CRC and IBD, but the enriched pathways differed between diseases (Figure 3 and Figure 4). OTGs of *S. oralis* were enriched in purine metabolism in CRC, while in two subtypes of IBD, they primarily participated in quorum sensing. Similarly, *G. morbillorum* showed different pathways in CRC and IBD-CD. While the former was enriched in peptidoglycan biosynthesis, the latter emphasizes teichoic acid biosynthesis. These findings underscored the key role of *S. oralis* and *G. morbillorum* in the pathogenesis of gastrointestinal disorders, suggesting potential adaptations of metabolic strategies in different diseases. A comparison of the top 20 COGs in CRC and IBD revealed a notable overlap (Table 2). Specifically, seventeen COGs were shared between the two IBD subtypes, and nine of them were also identified in CRC. These shared COGs primarily participate in cell wall biosynthesis, signal transduction, and DNA-binding response, suggesting potential implications for bacterial survival and adaptations during translocation events.

## 4. Discussion

Colorectal cancer (CRC) and inflammatory bowel disease (IBD) are prevalent gastrointestinal conditions, with emerging evidence suggesting a link between their development and alterations in the gastrointestinal microbiota [8,14,15,56]. However, the precise nature of this association, particularly regarding the influence of oral bacteria on gastrointestinal disorders, remains an area of active investigation [15,24]. In this study, we explore for the first time the association between the translocation of oral pathogens and gastrointestinal cancers at the genetic level to provide new insights into understanding microbial interactions in the oral–gut axis.

Previous studies have noted an abnormal enrichment of oral bacteria in individuals with IBD and CRC, implying a potential connection between the translocation of oral pathogens and gastrointestinal disorders [18,20,21]. Noteworthy oral resident pathogens, including *G. morbillorum*, *Veillonella* spp., and *Streptococcus* spp., have been reported to cause opportunistic infections, like endocarditis, sepsis, and pneumonia in immunocompromised patients, potentially resulting from the translocation of oral bacteria to the bloodstream or other organs [57,58,59]. A substantial number of OTGs identified in these bacteria provides additional evidence for such translocation events. Moreover, we brought to light a notable increase in the abundance of a Gram-negative bacterium named *L. adecarboxylata* in the gastrointestinal microbiota of CRC patients. This bacterium has been recognized as a novel conditional pathogen that may cause various microbial infections, such as sepsis, peritonitis, pneumonia, and urinary tract infections, particularly in immunocompromised individuals [60]. Notably, the OTGs of this bacterium were significantly enriched in 15 metabolic pathways, implying a crucial role in the modulation of the gut microbiome.

We found a large number of shared OTGs in both IBD and CRC, primarily involved in signal transduction and cell wall biosynthesis. Although the mechanism of oral bacteria translocation in the pathogenesis of gut diseases remains unclear, studies have suggested that specific oral pathogens can modulate epithelial innate immune responses through signal transduction systems [61]. This allows them to escape from host immune reactions, influencing the epithelial barrier function by altering the expression and distribution of cell–cell interactions and ultimately impacting the progression of diseases. Moreover, cell wall components of oral bacteria can also modulate host cell signal systems by binding to tumor cell receptors and triggering inflammation [62]. Typical examples include *Porphyromonas gingivalis* and *Helicobacter pylori*. The peptidoglycan of the former can bind to the TLR2 receptor on the surface of tumor cells and induce inflammatory responses, and the latter activates nuclear factor kappa-light-chain-enhancer of activated B cells (NF-κB) signaling in host cells with the help of synergistic effects of the cytotoxin-associated gene A (CagA), vacuolating cytotoxin A (VacA), and peptidoglycan, which are involved in the development of gastric cancer [63,64]. It is important to note that various factors—such as gastrointestinal disorders, like IBD and CRC, treatments, including antibiotics and immunosuppressive drugs, and oral and dental diseases—may contribute to the translocation and transmission of microbiota between the oral cavity and the intestines [18,65]. The complex interplay of these factors is still an area of active research, and the causal relationship between these conditions and microbial translocation events remains to be fully established and further validated. Moreover, understanding when commensal microorganisms or oral residents transition into a pathogen or pathobiont in the gut remains a significant challenge. This threshold is likely influenced by host immune status, microbial interactions, and environmental conditions, necessitating further study to fully elucidate these mechanisms and their implications for gut health and disease progression.

A comprehensive and accurate database is essential for functional annotation, which is a key step for studying the structure and function of microbial communities. However, information retrieval and analysis from commonly used microbial databases, such as the reference sequence (RefSeq) database at NCBI [66], GenBank [67], and the Integrated Microbial Genomes with Microbiome Samples (IMG/M) [68], are challenging due to their overwhelming size and complexity. In this study, we chose ChocoPhlAn3 as the pangenome database for gene-level functional analysis, which is a well-organized collection of representative genomes from different habitats and has been widely used for metagenomic studies. Despite the limitations of this database in terms of genome diversity, we consider it a reasonable compromise between data volume and genome quality. Moreover, while studies at the gene level yield more detailed information compared to those at the species level, they face the challenge of handling large volumes of data. The increased data volume not only raises computational complexity and time but also constrains the application of many commonly used differential abundance analysis methods. To address this problem, we strictly controlled the number of genes and annotation quality by adjusting HUMAnN3 parameters and followed the best practices and recommendations of previous studies by employing diverse differential abundance analysis methods to identify OTGs. However, we acknowledge the potential issue of false positives inherent in these methods and admit that they may not be optimal solutions. Future work is expected to explore and apply more advanced and suitable data analysis methods to enhance the efficiency and accuracy of gene-level analysis.

Another limitation of our study is that the characterization of the gut microbiota was based solely on fecal samples. While fecal microbiota profiling is a widely used method, it may not fully capture the microbial landscape of the entire digestive tract. Ideally, a comprehensive analysis would also include saliva (with gingival crevicular fluid), blood, and urine samples, as differences in microbial communities exist throughout the digestive tract, even in healthy individuals. Additionally, due to restrictions on human-related metadata in public databases, some important information, such as the time of sample collection, host gender, and age, was difficult to obtain. Future studies should consider incorporating these additional sample types and metadata to gain a more complete understanding of the gut microbiota and its role in gastrointestinal health and disease.

In conclusion, we have identified potential oral translocation genes associated with colorectal cancer and inflammatory bowel disease using multiple differential abundance analysis methods. These genes were enriched in biological functions and metabolic pathways that may be relevant to bacterial survival, adaptation, and communication within the gut environment. However, our findings should be viewed as exploratory, and the generalizability of these results across different individuals remains uncertain due to the diverse immune responses that can either counteract or exacerbate microbial translocation. Further investigations are needed to validate these associations and fully understand the implications of oral–gut pathogen transfer in various gastrointestinal conditions.

## Figures and Tables

**Figure 1 microorganisms-12-02086-f001:**
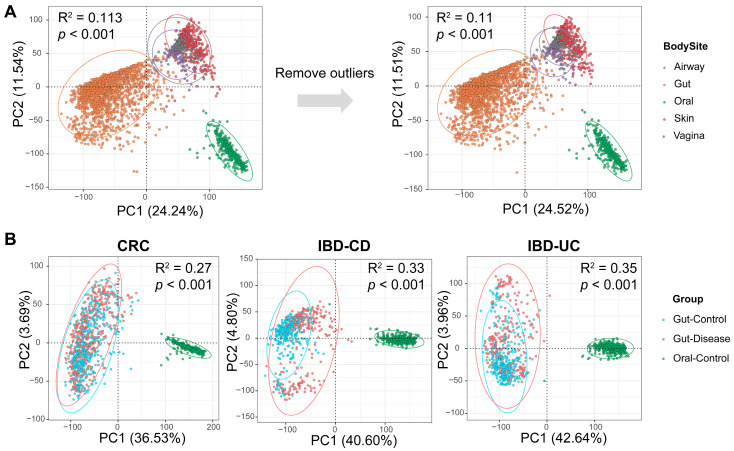
Principal component analysis (PCA) of samples from different body sites. (**A**) PCA plot of samples from five major body sites before and after outlier removal. Clustering was performed based on the CLR-transformed species relative abundance data of each sample. (**B**) PCA plot of oral samples and gut samples of disease subjects and healthy controls. R^2^ and *p*-values were calculated by PERMANOVA.

**Figure 2 microorganisms-12-02086-f002:**
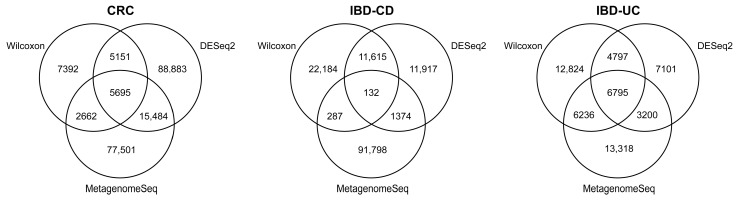
Venn diagram of the number of translocation genes identified by three differential abundance analysis methods. Numbers in parentheses represent the total number of candidate translocation genes identified by each method. CRC, colorectal cancer; IBD, inflammatory bowel disease; CD, Crohn’s disease; UC, ulcerative colitis.

**Figure 3 microorganisms-12-02086-f003:**
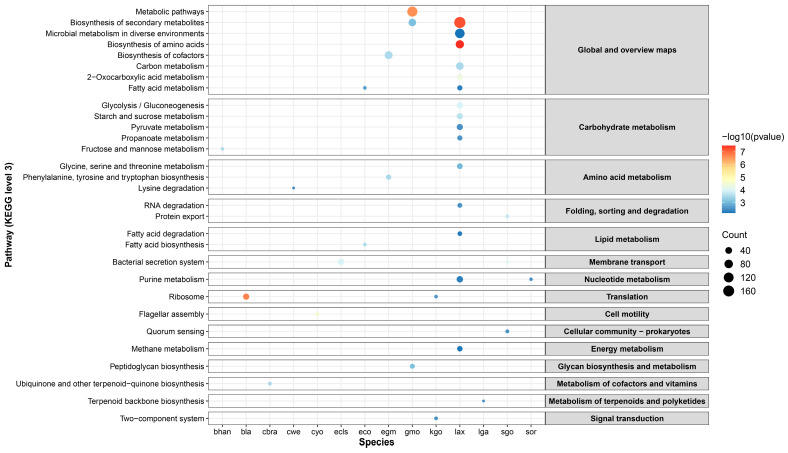
Pathway enrichment of OTGs in CRC. The x-axis indicates the abbreviations of different species and the y-axis shows the enriched level 3 (**left**) and level 2 (**right**) KEGG pathways. The bubble color represents the −log10(*p*-value) and the size indicates the number of OTGs. Abbreviations: bhan, *Blautia hansenii*; bla, *Bifidobacterium animalis*; cbra, *Citrobacter braakii*; cwe, *Citrobacter werkmanii*; cyo, *Citrobacter youngae*; ecls, *Enterobacter cloacae complex*; eco, *Escherichia coli*; egm, *Elizabethkingia bruuniana*; gmo, *Gemella morbillorum*; kgo, *Kluyvera georgiana*; lax, *Leclercia adecarboxylata*; lga, *Lactobacillus gasseri*; sgo, *Streptococcus gordonii*; sor, *Streptococcus oralis*.

**Figure 4 microorganisms-12-02086-f004:**
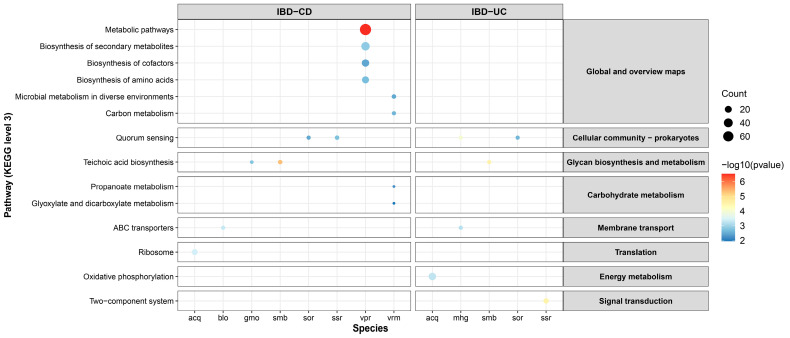
Functional annotation and pathway enrichment of OTGs in CD and UC. Abbreviations: acq, *Actinomyces* sp. oral taxon 414; blo, *Bifidobacterium longum*; gmo, *Gemella morbillorum*; smb, *Streptococcus mitis*; sor, *Streptococcus oralis*; ssr, *Streptococcus salivarius*; vpr, *Veillonella parvula*; vrm, *Veillonella rodentium*; mhg, *Megamonas hypermegale*.

**Table 1 microorganisms-12-02086-t001:** Summary of datasets used in this study. CRC, colorectal cancer; IBD, inflammatory bowel disease; CD, Crohn’s disease; UC, ulcerative colitis. The acronyms PRJDB, PRJEB, and PRJNA refer to project identifiers in the DDBJ, ENA, and NCBI databases, respectively.

Dataset ID	Total Samples	Control Samples	Disease Samples	Reference
Gut-CRC (8)	861	438	423	
PRJDB4176	78	40	38	[28]
PRJEB10878	128	54	74	[32]
PRJEB27928	81	60	21	[20]
PRJEB6070	60	20	40	[33]
PRJEB7774	109	63	46	[34]
PRJNA429097	193	95	98	[35]
PRJNA447983	49	21	28	[25]
PRJNA731589	163	85	78	[36]
Gut-IBD (5)	808	334	474	
PRJEB1220	232	115	15 (CD), 102 (UC)	[37]
PRJEB15371	85	39	46 (CD)	[38]
PRJEB50555	224	56	83 (CD), 85 (UC)	[39]
PRJNA398089	137	35	61 (CD), 41 (UC)	[40]
PRJNA793776	130	89	41 (CD)	[41]
Oral (1)	363			
PRJNA48479	363			[42]
Airway (2)	149			
PRJNA275349	18			[43]
PRJNA48479	131			[42]
Skin (3)	594			
PRJNA46333	194			[44]
PRJNA604820	200			[45]
PRJNA763232	200			[46]
Vagina (2)	138			
PRJNA275349	3			[43]
PRJNA48479	135			[42]

**Table 2 microorganisms-12-02086-t002:** Top 20 COGs of OTGs in CRC and IBD groups.

COG ID	Annotation	COG Symbol	Disease Type
COG0438	Glycosyltransferase is involved in cell wall biosynthesis	RfaB	CRC/IBD-CD/IBD-UC
COG0456	Ribosomal protein S18 acetylase RimI and related acetyltransferases	RimI	CRC/IBD-CD/IBD-UC
COG0463	Glycosyltransferase is involved in cell wall biosynthesis	WcaA	CRC/IBD-CD/IBD-UC
COG0642	Signal transduction histidine kinase	BaeS	CRC/IBD-CD/IBD-UC
COG0745	DNA-binding response regulator, the OmpR family, contains REC and a winged-helix (wHTH) domain	OmpR	CRC/IBD-CD/IBD-UC
COG1309	DNA-binding protein, the AcrR family, includes nucleoid occlusion protein SlmA	AcrR	CRC/IBD-CD/IBD-UC
COG2197	DNA-binding response regulator, the NarL/FixJ family, contains REC and HTH domains	CitB	CRC/IBD-CD/IBD-UC
COG2814	Predicted arabinose efflux permease AraJ, the MFS family	AraJ	CRC/IBD-CD/IBD-UC
COG4974	Site-specific recombinase XerD	XerD	CRC/IBD-CD/IBD-UC
COG0596	2-succinyl-6-hydroxy-2,4-cyclohexadiene-1-carboxylate synthase MenH and related esterases, alpha/beta hydrolase fold	MenH	IBD-CD/IBD-UC
COG1131	ABC-type multidrug transport system, ATPase component	CcmA	IBD-CD/IBD-UC
COG1132	ABC-type multidrug transport system, ATPase, and permease component	MdlB	IBD-CD/IBD-UC
COG1136	ABC-type lipoprotein export system, ATPase component	LolD	IBD-CD/IBD-UC
COG1396	Transcriptional regulator, which contains XRE family HTH domain	HipB	IBD-CD/IBD-UC
COG1846	DNA-binding transcriptional regulator, the MarR family	MarR	IBD-CD/IBD-UC
COG4585	Signal transduction histidine kinase ComP	ComP	IBD-CD/IBD-UC
COG4932	Clumping factor A-related surface protein, the MSCRAMM (microbial surface components recognizing adhesive matrix molecules) family, DEv-IgG fold	ClfA	IBD-CD/IBD-UC
COG1609	DNA-binding transcriptional regulator, the LacI/PurR family	PurR	CRC/IBD-CD
COG2801	Transposase InsO and inactivated derivatives	Tra5	CRC/IBD-UC
COG0582	Integrase/recombinase, which includes phage integrase	FimB	CRC
COG0583	DNA-binding transcriptional regulator, the LysR family	LysR	CRC
COG0697	Permease of the drug/metabolite transporter (DMT) superfamily	RhaT	CRC
COG1028	NAD(P)-dependent dehydrogenase, the short-chain alcohol dehydrogenase family	FabG	CRC
COG2207	AraC-type DNA-binding domain and AraC-containing proteins	AraC	CRC
COG3121	P pilus assembly protein, chaperone PapD	FimC	CRC
COG3188	Outer membrane usher protein FimD/PapC	FimD	CRC
COG3210	Large exoprotein involved in heme utilization or adhesion	FhaB	CRC
COG3539	Pilin (type 1 fimbrial protein)	FimA	CRC
COG0457	Tetratricopeptide (TPR) repeat	TPR	IBD-CD
COG2226	Ubiquinone/menaquinone biosynthesis C-methylase UbiE/MenG	UbiE	IBD-CD
COG0515	Serine/threonine protein kinase	SPS1	IBD-UC
COG0789	DNA-binding transcriptional regulator, the MerR family	SoxR	IBD-UC

## Data Availability

Oral metagenomic sequencing data and related metadata were downloaded from the Human Microbiome Project Data Portal (https://portal.hmpdacc.org/ (accessed on 30 October 2022)). Sequencing data of samples from the intestines and other body sites were also downloaded from public databases, including the Sequence Read Archive (SRA, https://trace.ncbi.nlm.nih.gov/Traces/sra/ (accessed on 16 November 2022)), the European Nucleotide Archive (ENA, https://www.ebi.ac.uk/ (accessed on 16 November 2022)), and the DNA Data Bank of Japan (DDBJ, https://www.ddbj.nig.ac.jp/ddbj/index-e.html (accessed on 16 November 2022)).

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
