# Peer review of "Oral Microbial Translocation Genes in Gastrointestinal Cancers: Insights from Metagenomic Analysis"

_microorganisms, 2024, doi:10.3390/microorganisms12102086_

Round 1

Reviewer 1 Report

Comments and Suggestions for Authors

Review 

Journal Microorganisms (ISSN 2076-2607)

Manuscript ID microorganisms-3240252

Type Article

Title Oral Microbial Translocation Genes in Gastrointestinal cancers: Insights from Metagenomic Analysis

Authors Linqi Wang , Qinyu Wang , Yan Zhou *

Section Microbiomes

  This article is based on studies at the bacterial species level that have proven the ability of certain oral pathogens to migrate to the intestine by translocation. The study uses genetic evidence to support this mechanism.   Among bacteria, the presence of the oral translocation genes of Leclercia adecarboxylata, an opportunistic pathogen, whose abundance is significantly high in the intestinal microbiota of colorectal cancer patients. From this discovery for the first-time evidence at the genetic level of the transfer of oral pathogens to the intestine is put forward by the article. This discovery could offer new perspectives in understanding the role of oral pathogens in the development of gastrointestinal cancers.   The article must point out certain obvious facts.      There is no evidence today that the oral microbiota is at the origin of CRC, however there is some that suggests that the intestinal microbiota is strongly associated with the development and progression of the CRC. The actual origin of CRC remains multifactorial.  Several changes in the gut microbiome accompany the progression of CRC.   It appears that gut microflora profiles associated with other factors...age, diet, family history, body mass index and geographical location are currently being studied. The benefits of dietetics for

 reshaping the intestinal microbiota and helping prevent and treat CRC seem interesting. Some authors propose diagnostic tests based on the intestinal microbiota (biomarkers) that can provide reliable and precise identification of risk factors for developing CRC[Pandey et al.,2023].

  For more, the precise mechanism by which an oral infection, such as periodontal disease, contributes to the pathogenesis of extra-oral diseases remains poorly understood in vivo.  One hypothesis: Periodontitis leads to an expansion of oral pathobionts. These oral pathobionts can be ingested and migrate to the intestine, where they activate the inflammasome in colonic mononuclear phagocytes, which can trigger inflammation.  Another hypothesis comes from Th17 cells reactive to oral pathobionts in the oral cavity. These Th17 cells sensitive to oral pathobionts migrate to the intestine. When in the intestine, Th17 cells of oral origin can be activated by translocated oral pathobionts and cause the development of colitis.  Thus, this scenario shows how periodontal disease provides the intestine with pathobionts but also pathogenic T cells. [Kilamoto et all.,2020].      

Some details to correct on the form and the background...

Line 38: why repetition of Streptococcus spp? is it rather Staphylococcus spp?

Line 84 : is it FastQC A Quality Control tool for High Throughput ...?

Line92 :  more details regarding the acronyms PRJDB, PRJEB, PRJNA ?

Heterogeneity of studies, protocols (different Next-generation Sequencing studies, sample, moment of the day..)

references ( 48 to 55) are basing on faecal sample uniquely  not on blood or other ?

  Line 153 :  The hypothesis of the possibility of translocation in a healthy individual between the microbiomes of the digestive tract and the oral cavity and vice versa must be raised.  Especially since microbial circulation can also exist from the digestive tract to the oral cavity in people who regurgitate or who present a translocation following benign inflammation of the digestive tract.

Line 202 : reference of this sentence.

Line 226 : translate COG and KEGG…

Line 240 :  the difficulty is to know “the potential role” of the different intestinal and oral micro-organisms.  Furthermore, the difficulty is to know from when a commensal microorganism or a foreign oral resident becomes (from what threshold) a pathogen or a pathobiont in the intestine?

Line 244 : Reference at the end of paragraphe.

Line 325 : The conclusion of article reference (7) is less categorical. Indeed, the authors specify that the microbial influence on translational approaches to the prevention and treatment of CRC remains to be demonstrated. As a preliminary step, it is essential to understand the underlying biology of microbially mediated CRC before considering diagnostics or therapeutic strategies based on the use of microbiota to manipulate the tumor microenvironment.

Line 331 :   a reference is necessary following this sentence. In addition, several questions may arise following this observation... can IBD or CRC or different treatments (ATB immunosuppressives) or oral and dental diseases (or all) be the cause of translocation, transmission  (between the mouth and the intestine or between the intestinal and oral microbiota).

Line 335 : HIV is another recent pathology incrimined ...

Li S, Su B, Wu H, He Q, Zhang T. Integrated analysis of gut and oral microbiome in men who have sex with men with HIV Infection. Microbiol Spectr. 2023 Dec 12;11(6):e0106423. doi: 10.1128/spectrum.01064-23. Epub 2023 Oct 18. PMID: 37850756; PMCID: PMC10714972.

Line 354: NF-kB in full letters ...

Line 355: idem cag A and VacA in full ..

Line 389 :  the characterization of the digestive microbiota based solely on feces is insufficient. Indeed actually if you want characterize gut microbiota is necessary to analyze saliva (with GCF), fecal sample, blood and urine samples. Knowing that differences between healthy individuals exist throughout the digestive tract.

The article suffers from the lack of a recent bibliography on this subject as...

Koliarakis I, Messaritakis I, Nikolouzakis TK, Hamilos G, Souglakos J, Tsiaoussis J. Oral Bacteria and Intestinal Dysbiosis in Colorectal Cancer. Int J Mol Sci. 2019 Aug 25;20(17):4146. doi: 10.3390/ijms20174146. PMID: 31450675; PMCID: PMC6747549.

Wang Z, Dan W, Zhang N, Fang J, Yang Y. Colorectal cancer and gut microbiota studies in China. Gut Microbes. 2023 Jan-Dec;15(1):2236364. doi: 10.1080/19490976.2023.2236364. PMID: 37482657; PMCID: PMC10364665.   Kitamoto S, Nagao-Kitamoto H, Jiao Y, Gillilland MG 3rd,  Hayashi A, Imai J, Sugihara K, Miyoshi M, Brazil JC, Kuffa P, Hill BD, Rizvi SM, Wen F, Bishu S, Inohara N, Eaton KA, Nusrat A, Lei YL, Giannobile WV, Kamada N. The Intermucosal Connection between the Mouth and Gut in Commensal Pathobiont-Driven Colitis. Cell. 2020 Jul 23;182(2):447-462.e14. doi: 10.1016/j.cell.2020.05.048. Epub 2020 Jun 16. PMID: 32758418; PMCID: PMC7414097.  

Pandey H, Tang DWT, Wong SH, Lal D. Gut Microbiota in Colorectal Cancer: Biological Role and Therapeutic Opportunities. Cancers (Basel). 2023 Jan 30;15(3):866. doi: 10.3390/cancers15030866. PMID: 36765824; PMCID: PMC9913759.

Concerning Lecleria adecarboxylata..

Dotis J, Kondou A, Karava V, Sotiriou G, Papadopoulou A, Zarras C, Michailidou C, Vagdatli E, Printza N. Leclercia adecarboxylata in Peritoneal Dialysis Patients: A Systematic Review. Pediatr Rep. 2023 Apr 25;15(2):293-300. doi: 10.3390/pediatric15020025. PMID: 37218925; PMCID: PMC10204424.

Shaikhain T, Al-Husayni F, Al-Fawaz S, Alghamdi EM, Al-Amri A, Alfares M. Leclercia adecarboxylata Bacteremia without a Focus in a Non-Immunosuppressed Patient. Am J Case Rep. 2021 Mar 30;22:e929537. doi: 10.12659/AJCR.929537. PMID: 33782375; PMCID: PMC8019838.

Tan R, Yu JQ, Wang J, Zheng RQ. Leclercia adecarboxylata infective endocarditis in a man with mitral stenosis: A case report and review of the literature. World J Clin Cases. 2022 Oct 16;10(29):10670-10680. doi: 10.12998/wjcc.v10.i29.10670. PMID: 36312476; PMCID: PMC9602224.

Concerning HIV infection

Li S, Su B, Wu H, He Q, Zhang T. Integrated analysis of gut and oral microbiome in men who have sex with men with HIV Infection. Microbiol Spectr. 2023 Dec 12;11(6):e0106423. doi: 10.1128/spectrum.01064-23. Epub 2023 Oct 18. PMID: 37850756; PMCID: PMC10714972.

Conclusion :  the article presents the synthesis of several microbial analyzes of feces (IBD and CRC) of saliva (oral), airway, skin and vagina. Basing on difference of abundance between oral and gut samples with three statistical methods the article try to identify genes associated with translocation. However, the conclusion must be accompanied by several reservations in order to be acceptable. Indeed, it seems difficult to generalize the phenomenon of translocation to all individuals. To the extent that each healthy or sick individual presents their own immunity capable of countering or aggravating an inflammatory process.

 v

Reviewer 2 Report

Comments and Suggestions for Authors

Introduction

Line 37: The names of the bacteria Fusobacterium spp., Klebsiella spp., Streptococcus spp., Porphyromonas spp.

They should be put in italics

Figure 2: They should identify the X and Y axes, the scale in the figure is not clear, if there are units or how they are interpreted, they should be put.

Figures 4 and 5. The microorganisms cannot be seen, and the image is tiny and blurry, which makes it difficult to understand. The quality of the image must be improved.

The writing of the bacteria must comply with the taxonomic standards for these genera, with the first letter in capital letters and the species in all lowercase letters and they must be in italics. The first time it is named, it is written in full, then the capital letter of the genus is written, followed by a period and the species. It is not appropriate to write it this way. They must correct Streptococcus oralis (S. oralis)

This is an excellent work in which they have used previously reported public data. However, despite having such good data, the images do not merit the work. Additionally, they could do a more robust analysis with this data such as a Microbial network analysis, scatter plot, Volcano plot, or neural networks, this would greatly improve the results and confirm the data that you indicate here, which as you show is still not so clear.

I suggest complementing the analysis and strengthening the discussion, as well as indicating the limitations of the study.

Cordially,

Reviewer 3 Report

Comments and Suggestions for Authors

This manuscript aimed to discover oral translocation genes (OTGs) associated with colorectal cancer (CRC) and inflammatory bowel disease (IBD). The concept is interesting, but I do have some fundamental questions regarding how OTG are defined. Additionally, ALL figures are too blurry to read that prevents me from assessing the contents in full capacity. I believe the authors can address the questions and improve the manuscript.

Table 1: information regarding control and disease are missing for oral, airway, skin and vagina datasets.

Section 2.5: The basal bacterial abundance is much higher in gut than in oral. How do correct for different basal level when defining translocation genes? I would expect OTG to have much higher abundance in gut than oral in disease samples instead of no significance.

All figures are too blurry to read.

Section 3.1 and figure 1: Please also provide the pairwise comparison p value from PERMANOVA. 

Line 145-147: The authors included samples from respiratory tract, skin, and vagina to assess potential contamination. But the results from these sites are barely discussed. I think these samples are distractive to the main point and the authors should consider removing those samples.

Line 149-150: It would be informative to also compare disease oral samples with colorectal cancer, Crohn's disease, and ulcerative colitis gut samples.

The content of section 3.2 obstructs the flow of the manuscript. Section 3.2 can be incorporated into methods and Figure 2 can go to supplementary.

Line 202-204: The definition of OTG here is somewhat contradictory to what is described in method.

Line 213-215: The authors cannot comment on the sensitivity of the test, because the authors do not know whether these OTGs are true positive or not.

Line 235: italicize the bacteria name.

Table 2: Gene name should be COG symbol. These two are not the same.

Data Availability: Authors should provide a full list of taxonomic profile from all samples. Same with a list of COG and KEGG pathways.

Round 2

Reviewer 1 Report

Comments and Suggestions for Authors Thank you to the authors for making all the clarifications and modifications to their article as well as the necessary reserve. Line 135: reference at the end of this paragraph is necessary to support the approach.ions.

Reviewer 3 Report

Comments and Suggestions for Authors

The authors have greatly improved clarity and stringency of the manuscript. I am extremely satisfied.